# Characterization of the Structure and Transport Properties of Alginate/Chitosan Microparticle Membranes Utilized in the Pervaporative Dehydration of Ethanol

**DOI:** 10.3390/polym12020411

**Published:** 2020-02-11

**Authors:** Gabriela Dudek, Przemysław Borys, Anna Strzelewicz, Monika Krasowska

**Affiliations:** Department of Physical Chemistry and Technology of Polymers, Faculty of Chemistry, Silesian University of Technology, Strzody 9, 44-100 Gliwice, Poland; Przemyslaw.Borys@polsl.pl (P.B.); Anna.Strzelewicz@polsl.pl (A.S.); Monika.Krasowska@polsl.pl (M.K.)

**Keywords:** hybrid membranes, pervaporation, fractal analysis, random walk, transport model

## Abstract

The structure and transport properties of alginate/chitosan microparticle membranes used in ethanol dehydration processes were investigated. The membranes were characterized based on images obtained from high-resolution microscopy. The following parameters were determined: the observed total amount of void space, the average size of the void domains, their length and diameter, the fractal dimension, and the generalized stochastic fractal parameters. The total amount of void space was determined to be between 54% and 64%. The average size of the void domains is smaller for alginate membranes containing neat (CS) and phosphorylated (CS-P) chitosan particles when compared to those membranes filled with glycidol-modified (CS-G) and glutaraldehyde crosslinked (CS-GA) chitosan particles. Furthermore, the transport of ethanol and water particles through the studied membranes was modelled using a random walk framework. It was observed that the results from the theoretical and experimental studies are directly correlated. The smallest values of water to ethanol diffusion coefficient ratios (i.e., 14) were obtained for Alg (sodium alginate) membranes loaded with the CS and CS-P particles, respectively. Significantly larger values (27 and 19) were noted for membranes filled with CS-G and CS-GA particles, respectively. The simulation results show that the size of channels which develop in the alginate matrix is less suited for ethanol molecules compared to water molecules because of their larger size. Such a situation facilitates the separation of water from ethanol. The comparison of the structural analysis of the membranes and random walk simulations allows one to understand the factors that influence the transport phenomena, in the studied membranes, and comment on the effect of the length, diameter, number of channels, and variations in the pore diameters on these transport parameters.

## 1. Introduction

Pervaporation is a novel and rapidly developing membrane technology [1,2,3] which is considered effective and low energy consuming compared to other separation methods of near boiling mixtures like the dehydration of organic compounds, the recovery of organic compounds from water, and the separation of organic-organic mixtures. Moreover, pervaporation does not require the use of additional substances which contribute to the contamination of the mixture. Currently, approximately one hundred pervaporation units are operating worldwide and most of them dehydrate different solvents, such as ethanol, isopropanol, and tetrahydrofuran [4,5,6]. 

Current research in the pervaporation field is focused on the development of new membrane materials and optimization of the pervaporation process parameters. To advance in both areas, it is necessary to characterize the membrane structure and understand the mass transport processes that are involved in the pervaporation process. One of the methods to study the structure and morphology of the membranes is the fractal analysis. Using the fractal dimension *D_F_* and the generalized fractal dimension *Dq*, it is possible to quantify the structure and morphology of self-similar objects. In our previous work [7], we have shown that there is a strong relationship between the degree of multifractality Δ*D* and the separation efficiency. Comparing the trends in the changes of the Pervaporation Separation Index (PSI) and the corresponding Δ*D* values, we have noted, that the maximum PSI was always reached for the lowest values of Δ*D*. The lower value of Δ*D* is related to a more homogeneous and self-similar membrane structure. In order to predict the membrane’s performance and to design membranes for specific applications, a fundamental understanding of the transport phenomena is required. The classical description of the transport through permselective membranes involves four successive steps: diffusion of the component through the liquid boundary layer to the membrane surface, sorption/diffusion into the membrane, transport through the membrane, and, finally, diffusion through the vapor phase boundary layer into the bulk of the permeate. 

Most of the models described in the literature concern the sorption, whereas the amount of diffusion models is limited. The development of sorption models has changed from empirical models with low complexity to theoretical models with high complexity. The nature of modern sorption models is mainly semi-empirical and the diffusion models is mostly empirical [8]. The selection of an appropriate model is determined by the requested application [8]. Empirical models describe the process based on the measurement data, without taking into justification any physio-chemical relations. Cojocaru et al. [9] used factorial modeling and analysis, and the desirability function to solve the conflicting relationship between the two pervaporation parameters, the total permeate flux and selectivity, to obtain high values of both. The model was tested for water/acetonitrile and water/ethanol mixtures. Hafrat et al. [10] developed a model based on diffusion equations, the balance of mass and energy. Simulation models were designed using a simple program with Scilab. The aim of this model was to improve ethanol production and optimize energy consumption.

Theoretical models use molecular process parameters including their physical relationships. These molecular parameters are derived from thermodynamic and physicochemical relations. Shieh et al. [11] used a pseudo-phase-change solution-diffusion model for pervaporation of binary mixtures through a membrane. As an example, the authors studied the experimental data of the hexane/heptane/polyethylene pervaporation system—available in the literature. They noted a good agreement between the observed and calculated membrane efficiency in terms of the effects of feed pressure, permeate pressure, and feed concentration.

Phenomenological/semi-empirical models combine theoretical features with empirical approaches. Phenomenological models are mainly based on an approved theoretical background. Consequently, the number of required experiments to get a quantitative reasonable model can be minimized without reducing the quality of the approach. Over the years, several semi-empirical related to the solution-diffusion model have been developed. One of them, the Meyer-Blumenroth model [12], extended the initial solution-diffusion model by introducing a coupling effect for up to three feed components. The driving force in this model is the fugacity, instead of the concentration, that is based on the activity present in the initial solution-diffusion model. Another alternative model to predict fluxes related to the solution-diffusion has been developed by Klatt. In this model, the flux is described as a function of the permeability and the difference in the chemical potential between the feed and the permeate side [13]. Vier [14] used a further approach by including coupling in the solution-diffusion model and dividing the permeability into permeability functions. 

Further research on pervaporation models gives the opportunity to consider new combinations of models by combining known models with the free volume theory, permeate pressure drop, support layer, concentration polarization and so on. Future work is likely to converge on the models based on the molecular simulations. Such models could provide information on membrane diffusion and sorption behavior before polymer synthesis [8].

The presented paper is focused on formulating a relationship between the structure, transport, and the resulting pervaporation performance. In order to achieve this goal, the alginate/chitosan microparticle membranes were investigated for the ethanol dehydration process. From our previous research [15], the alginate membranes filled with various chitosan particles display excellent properties for the dehydration of ethanol by pervaporation. Furthermore, the structure of the membranes is very specific and is not observed in other hybrid membranes, being filled with inorganic fillers. For this reason, the characterization of such membranes was completed by scanning electron microscopy. The transport parameters were evaluated based on pervaporation experiments and were compared to the results obtained from transport modelling and fractal analysis.

## 2. Experimental

### 2.1. Membrane Preparation

A 1.5 wt % sodium alginate solution was prepared by dissolving an appropriate amount of sodium alginate powder in deionized water. This solution was mixed with a suitable portion of chitosan (CS) or modified CS microparticles, such as phosphorylated chitosan (CS-P), glycidol-modified chitosan (CS-G), or glutaraldehyde crosslinked chitosan (CS-GA), to obtain the required filler concentrations, namely 5, 10, 15, and 25 wt %. The solutions were then cast onto Petri dishes and placed on a levelled plate and evaporated to a dry state at 60 °C. After 24 h, the membranes were crosslinked using calcium chloride. The pristine Alg membrane was prepared in the same manner as above, except for the addition of chitosan particles. The membrane thickness was measured using a waterproof precise coating thickness gauge MG-401 ELMETRON (Zabrze, Polska), and was estimated as the mean value of at least 10 measurements at different points. For all membranes, the thickness was 30.0 ± 2.0 μm.

### 2.2. Preparation of Modified Chitosan Particles

The preparation of modified chitosan particles i.e., glycidol-modified chitosan (CS-G), glutaraldehyde crosslinked chitosan (CS-GA) and phosphorylated (CS-P) chitosan particles was completed as previously outlined [15]. CS-G were prepared by mixing 200 mL of 3 wt % chitosan in 2 vol % acetic acid solution. The resultant CS-G particles were further suspended in 250 mL of deionized water, oxidized with 70 mL of 0.16 M sodium periodate (NaIO_4_), and slightly stirring for 2 h at room temperature. CS-GA particles were synthesized by adding 0.01 M NaOH drop-wise to the CS solution. After reaching a pH of 5.6, 1.4 mL of 50 wt % glutaraldehyde solution was added to the solution. Next, 2 M NaOH was added drop-wise into the liquid phase for 3 h until a dark brown suspension was produced, and the final pH reached 7. CS-P particles were prepared by adding 20 g of chitosan, 100 g of urea, and 20 g of 100 % orthophosphoric acid into 200 mL of dimethylformamide.

### 2.3. Characterization of Membrane Morphology

Surface characterization was completed using a Phenom Pro X SEM microscope (Eindhoven, The Netherlands). On the basis of the binarized membrane images obtained from the microscope, a fractal analysis concentrating on the fractal dimension *D_F_* and generalized fractal dimension *D_q_* were completed. These parameters allow one to quantify the structure and morphology of self-similar objects and is commonly used [16,17,18,19,20,21,22].

For the self-similar sets the number of nonempty coverings N(ε) scales with the size of the covering box element ε by:(1)$$N\left(\varepsilon \right)\propto \;\varepsilon^{-D_{F}}$$
where $D_{F}$ is a fractal dimension. Taking the logarithm under the limit ε→0, gives:(2)$$D_{F}=\underset{\varepsilon \to 0}{\lim}\frac{\ln\left(N\left(\varepsilon \right)\right)}{\ln\left(\frac{1}{\varepsilon }\right)}$$

The fractal dimension describes the object using a single number. Since various objects can have the same fractal dimension, to distinguish them the generalized fractal dimension is introduced:(3)$$D_{q}=\frac{1}{q-1}\underset{\varepsilon \to 0}{\lim}\frac{\ln\sum_{i=1}^{N\left(\varepsilon \right)}P_{i}^{q}}{\ln\varepsilon }$$
where *N*(*ε*) is the number of covering elements (boxes); *ε* is the size of the covering element (length of the edge); *q* is a real number (dimension index); and *P_i_* is the probability to find mass in the ith element (for example the number of mass pixels in an element divided by the total number of pixels in the element). The Box Counting Method (BCM) allows calculation of both the fractal dimension *D_F_* and its extension *D_q_*, termed the generalized fractal dimension [18,19,20,21,22]. For *q* = 0, the generalized fractal dimension *D_q_* is independent of the P_i_ distributions (any nonzero P_i_ raised to power 0 gives 1) and, therefore, equals *D_F_*. 

The fractal dimension exponents were determined with an accuracy of 0.05. The theoretical properties of the *D_q_* spectra have been previously described in detail [19]. The degree of multifractality ΔD is related to the deviation from a simple self-similarity and is the difference between the maximum and minimum generalized fractal dimension, which values are related to the least dense and most dense boxes in the phase space [23]:(4)$$\Delta D=D_{-\infty }-D_{\infty }$$

The f(α) ("multifractal”) spectrum is a Legendre transform of *D_q_*, defined as:(5)$$f\left(\alpha \right)=\alpha \tilde{q}\left(\alpha \right)-\tau_{\tilde{q}\left(\alpha \right)}$$
where:(6)$$\alpha =\frac{\partial \tau_{q}}{\partial \tilde{q}}$$
and
(7)$$\tau_{q}=\left(q-1\right)D_{q}=\underset{\varepsilon \to 0}{\lim}\frac{\ln\sum_{i=1}^{N\left(\varepsilon \right)}P_{i}^{q}}{\ln\varepsilon }$$
and $\tilde{q}$ is the dimension index. 

The full derivation of this transform has been previously discussed [19]. The f(α) spectrum delivers a better form of data presentation as it results in a curve with a maximum. This helps to show the characteristic dimensions of the curve (α_min_, α_max_, maximum). Both, the maximum and minimum dimensions, *D*_−__∞_ and *D*_∞_, depend on the extreme values of P_i_ and their difference ∆*D* (which is equivalent to Δα) and describes the range of values of P_i_. Thus, we have utilized ∆*D* (*D*_−__∞_ − *D*_∞_) or Δα (α_max_ – α_min_) as a measure of the self-similarity of the stochastic fractals. For an ideal/deterministic fractal, the value of ∆*D* equals 0. A large value of *D*_−__∞_ (α_max_) indicates the presence of very diluted regions. Conversely, a significant value of *D*_∞_ (α_min_) allows one to conclude that the dense regions are not very concentrated.

Excluding for the fractal parameters, the morphology of the binarized membrane image was characterized by the total amount of void space, which was defined as the ratio of the void space visible in the picture (black regions) to the total image area (porosity-like) and by the average size of the void domains. The size of the void domain was defined by the number of pixels in a black region surrounded by white pixels. Further parameters that characterize the membrane morphology were proposed whilst performing the simulation of the mass-transport through the membrane.

### 2.4. Pervaporation (PV) Experiments 

Pervaporation (PV) experiments were completed using the apparatus described by the authors of this script previously [24], under the same conditions (Figure 1). An aqueous solution of 96 vol % ethanol was used as a feed. The permeate was collected in a cold trap cooled with liquid nitrogen and the flux was calculated from the weight of the condensed permeate obtained at fixed time intervals. The composition of all three streams (feed, permeate, and retentate) was analyzed by gas chromatography using an Agilent Technologies 6850 gas chromatograph (Santa Clara, CA, USA) equipped with an autosampler, Elite-WAX column, and a flame ionization detector (FID). Nitrogen was used as a carrier gas and hydrogen as the fuel gas. To verify whether the results are statistically significant, three samples of each membrane type were fabricated, and for every membrane the PV experiment was repeated three times, giving nine independent measurements for each type of membrane. 

The permeation flux of component *i* was calculated using [25,26]:(8)$$J_{i}=\frac{m_{i}}{A\cdot t}$$
where *m*_i_ was the weight of component *i* in the permeate, A was the effective membrane area, and *t* was the permeation time to collect *m*_i_.

The diffusion coefficient was estimated using a previously described method [27]. This model takes into consideration the fact that the analyzed membranes are not empty before the measurement. It also considers the length of the tubing between the permeation cell and the cold traps. From this method the diffusion coefficient can be calculated using:(9)$$D=\frac{-l^{2}}{3L_{a}}$$
where: $L_{a}$ is the effective total Time Lag calculated as $L_{a}=L_{a}^{2}-6.5\cdot L_{a}^{1}$.

$L_{a}^{1}$—Time Lag for the tubing, s (for a line tangent to the mass curve at the inflection point)

$L_{a}^{2}$—the asymptotic Time Lag, s (for a line tangent to the mass curve on the asymptote for large time values)

l—the thickness of membrane, cm

The results acquired for the materials investigated in this study showed good repeatability and a standard deviation (SD) of less than 3%.

### 2.5. Transport Model

The transport of ethanol and water particles through alginate membranes filled with various chitosan particles was modelled using a random walk framework [28,29,30]. Using this model, the particles wander randomly over a discrete lattice (with a node spacing, δ), performing jumps in subsequent time steps, separated by time, τ. The jump distance and time step ratio are related to the velocity of random (thermal) motion of the particles by v= δ/ τ.

The random walk model parameters, δ and τ, are associated with physical properties of the system under consideration such as the mean free path and the time between collisions.

In this work, we use a three-dimensional random walk through the membrane volume, for each time step the permeating particle can move by +/− δ in one chosen direction. In such circumstances, the simulation parameters establish a transport process, characterized by the diffusion coefficient: [29]
(10)$$D=\frac{\delta^{2}}{6\cdot \tau }$$
where δ is the free path and τ is a time between collisions.

The transport through the membrane, which according to microscope images, displays a porous-like structure is modeled as a random wandering of the permeating particles within this porous structure. To establish this, a three-dimensional model of the membrane was required. We attempted to combine the microscope images of the membrane surface and the membrane cross-section. Unfortunately, the cross-sections were deformed during the preparation process, and we could not use these samples to model the structure in the bulk of the membrane.

To overcome this limitation, we decided to make use of the top-surface images, assuming that the deeper layers of the membrane are similar to this of the first layer. This is an assumption, that could not work, but eventually, the results show an agreement with the data, confirming that it is valid.

Within this assumption, we modeled the membrane as a set of subsequent layers, composed of translated versions of the top membrane surface, separated by an average pore diameter from each other. To obtain the diameter we had to perform additional structure analysis. Each void region of the image was analyzed to fit the largest possible circle, which diameter was then was assigned as representative for the analyzed region. Having such diameter, and void area from the previous structure analysis we could estimate the length of the channels and their area/diameter. The translation of the layers compared to the top membrane surface was required to prevent overlapping pores in subsequent layers (which is not expected in reality). Technically, the translation was completed using a 1024 × 1024 pixel wide image wrapped over the edges and translated randomly to the right and down, what escaped over the right edge, was put on the left, what escaped the bottom edge, was put on the top. The node spacing for a 1024 × 1024 × 1024 lattice corresponds to a microscope image of 26.7 nm.

## 3. Results and Discussion

### 3.1. Structure Analysis

The investigated alginate membranes filled with four various chitosan particles*,* neat (CS) phosphorylated (CS-P), glycidol-modified (CS-G), and glutaraldehyde crosslinked (CS-GA), were imaged in order to perform the structure analysis. The images of membranes were obtained using a Phenom Pro X SEM microscope (Eindhoven, The Netherlands). They were saved in the PNG format, which rendered no quality loss. The texture made by the chitosan particles and polymer matrix was observed at a magnification of ×10,000. To obtain the information about the space available for the particles transport; the pores within the membrane and their average size, we binarized the microscopic images to assign the attribution of the membrane pixel either to the polymer (if filled) or to void space (if empty, being part of a pore). The binarization was completed by means of the histogram analysis. After generating a histogram, the Gaussian peaks of the polymer signal and void signal were identified. Then, the cross-section of these curves was set as a threshold value. The resulting images were binary: 0 represented a void (black regions in Figure 2) and 1 represented a volume occupied by the polymer (white regions in Figure 2). The digitalized images of the membranes have a resolution of 1024 × 1024 pixels. One square micrometer corresponds to 6.12 × 6.12 pixels (one pixel corresponds to 26.7 nm × 26.7 nm).

The results of the structure analysis (the total amount of void space, the average size of void domains, the fractal dimension, and the degree of self-similarity) are set out in Table 1. The total amount of void space (porosity) is between 54% and 64%. The alginate membranes containing chitosan (CS) and phosphorylated chitosan (CS-P) particles have smaller domains than the membranes with glycidol-modified chitosan (CS-G) and the glutaraldehyde crosslinked chitosan (CS-GA) particles. This confirms the differences in the structure of the investigated fillers. Membranes with chitosan particles crosslinked by glutaraldehyde and modified by glycidol have larger void domains because such fillers extend the chain length between the monomers of chitosan, which concurrently leads to an increase in the free volumes in the membrane. 

The results of the fractal analysis show that the fractal dimensions of all investigated membranes are very similar to each other with overlapping uncertainty intervals. In addition, the degree of self-similarity Δα (homogeneity, the arrangement of morphological elements) has the same value for all membranes. In previous work [7], we have suggested that the arrangement of morphological elements may affect the transport properties. Here, however, the degree of self-similarity has the same value for all the investigated membranes, and thus cannot be used to explain their different transport properties.

Figure 3 provides the f(α) spectrum for the alginate/chitosan microparticle membranes. The analysis of the f(α) spectrum covers the width of the spectrum, the symmetry of branches, the altitude of maximum, placement of ends of the f(α) graph ends, and the presence of f(α) roots. 

The maximum height of f(α) corresponds to the fractal dimension. The highest value was found for the CS-P membrane structure and the lowest for CS-GA. The width of the spectrum, which reflects the self-similarity, is the same in all four cases and indicates a high degree of self-similarity of the membranes. This phenomenon can be explained because the membranes investigated in this manuscript contain organic filler which is compatible and easily dispersed in the polymer matrix. In our previous work [31,32,33], we have considered membranes with inorganic fillers. The results obtained now for hybrid membranes with organic filler are not in line with those of previous studies of organic/inorganic hybrid membranes. Due to the differences in the polymer matrix and the filler particles, the dispersion of inorganic filler is not as simple as in the case of organic filler. The filler dispersion in the polymer matrix and the self-similarity of the membrane affects its transport properties, which are better for membranes with a more self-similar structure.

### 3.2. Pervaporation Performance of Hybrid Alginate Membranes

Figure 4 presents diffusion coefficients for water and ethanol, while Figure 5 and Figure 6 compare the fluxes and Pervaporation Separation Index in a PV process through a pristine and three types of hybrid Alg membranes filled with modified CS particles (CS-P, CS-GA, and CS-G). 

It can be noted that due to the hydrophilic character of investigated membranes, the membrane surfaces prefer to absorb more water molecules and repel alcohol molecules, which is beneficial in the pervaporation dehydration process. Despite the differences between the chitosan derivatives which fill the alginate matrix, the ethanol diffusion coefficients remain similar. These values are also very similar to the ethanol diffusion coefficient estimated for pristine membrane. Huang et al. explained this phenomenon by suggesting hydrogen bonds forming between sodium alginate and chitosan, allowing easier transport of water over ethanol particles through the membrane [34]. Interestingly, the addition of various fillers has an effect on water diffusion coefficients. For all cases, the addition has a significant impact on this parameter, which is seen in comparison to the pristine alginate membrane. The largest diffusion coefficient value is observed for glutaraldehyde (CS-GA) and glycidol modified (CS-G) chitosan particles. This is due to crosslinking of chitosan by glutaraldehyde or modified by glycidol extends the chain length between monomers of chitosan more than in the case of pristine and phosphorylated chitosan particles. Consequently, this leads to an increase in the free spaces in the membrane. The largest values of water diffusion coefficients are obtained in the range of 5–15 wt % of chitosan particle content. Further addition of the filler particles to the matrix results in a decrease of the water diffusion coefficient. This phenomenon is explained by the membrane structure being more tightly packed up by the larger amount of chitosan particles and so there is less volume through which the penetrating particles can pass.

A similar relationship is noted for the flux (Figure 5). The flux through the pristine membrane is equal to 0.7 kg·m^−2^·h^−1^. The addition of the filler gives an increase of the flux up to 1.9 and 2.0 kg·m^−2^·h^−1^ for an alginate membrane containing glutaraldehyde (CS-GA) and glycidol modified (CS-G) chitosan particles. Contrary to the value of the water diffusion coefficient, there is observed a significant flux for the Alg membranes filled with phosphorylated chitosan particles. The observed increase in the flux could be attributed to the properties of the filling itself. The effect of the chitosan derivative particles on the membrane properties has been previously discussed [15,35]. Phosphorylation of chitosan introduces hydroxyl moieties present in phosphate groups, which affects the hydrophilicity of the membranes [36]. The introduction of such groups as phosphonic acid or phosphonate onto CS by the reaction of phosphorylating agent with the amino groups is known to increase the chelating properties of CS and, in consequence, induces better selectivity than for the membranes filled with CS, CS-GA, and CS-G particles (Figure 6). The nature of glycidol and glutaraldehyde groups, as well as the polymer chain packing by the filler, gives better permeability of ethanol molecules through the membrane. Therefore, a smaller PSI is obtained, compared to the Alg membranes filled with CS-P particles. 

### 3.3. Simulations of Random Walk

To initiate the simulation, we performed an analysis of the pore size distribution, observed in the binary images. The averages were different for each investigated structure (CS, CS-P, CS-GA, and CS-G) and equal to 24, 26, 36, 48 pixels, respectively. The void diameter distributions differed, displaying a (limited in range) power-law distribution of the diameter size (Figure 7). Such a power law dependency is critical in the analysis of the membrane transport parameters.

The expectations behind the simulations of a 3D porous network was that the permeation of the ethanol molecules should be more difficult than permeation of water molecules because the ethanol molecule is approximately twice as larger as water. Therefore, part of the porous percolation paths, that connect the feed side of the membrane to the output side, available for water, will be too narrow for the ethanol molecules.

The resolution of the microscopic images (fractions of a micrometer) is insufficient to model the porous network that directly affects the transport of the small molecules (water’s diameter is ~2.75Å). However, we may assume that the time to pass through the membrane (the First Passage Time [37]) should scale with the size of the membrane provided the structure of the pores can be assumed self-similar (and we have no better assumption in this modelling method). If we wish to determine the ratio of the First Passage Time for water and ethanol molecules, we can also analyze the ratio of First Passage Time for nano- or micrometer-sized particles with a similar mutual size relation, these would occupy several nodes of the simulation lattice. The absolute values of the First Passage Time will differ for each molecular system and for the larger system, but the ratio of the First Passage Time for a molecule compared to a molecule which is twice larger, is expected to be comparable on both size scales. 

If you consider a cubic self-similar membrane with a self-similarity scale factor equal to two i.e., the original cube can be composed out of 8 original cubes with a twice rescaled edge size. Each of the smaller cubes can be composed of 8 cubes, being twofold smaller, and such decomposition process can continue up to infinity. Now consider a micrometer-sized particle in front of the whole membrane, and increase its size by two. Some of the percolation paths become invalid for a larger particle compared to the smaller ones, the diffusion path through the less optimal pores becomes larger by a factor termed χ. We can relate this to an atomic-sized particle by considering such a particle in front of the membrane. A small, atomic-sized piece of the membrane in the intermediate vicinity of such a particle is just a rescaled version of the original membrane (as stated in the above assumption). Now if one enlarges the size of the atomic-sized particle by factor two. The transport through this small adjacent self-similar piece (not the whole membrane) will become hindered just the same way as it was for the micrometer-sized particle. The diffusion path through this piece elongates by the same factor χ as that of the micrometer-sized particle. Since the membrane is composed of such “tiny versions of the whole”, in each of these pieces, the diffusion path elongates in the same way. Therefore, the diffusion path in the whole membrane enlarges by the same ratio as for micrometer-sized particles. The First Passage Time is then related to the diffusion length by a power law, and this is why we hope for such scaling to hold for this system.

The power-law analysis is unfortunately limited in range even within the obtained data due to technical reasons: the resolution of the microscope images is finite and thus the pores for small diameters (in practice, below 7–8 pixels or 187–214 nm) are averaged out and their distribution becomes distorted (see the flattening of the distribution in Figure 6)). For large diameters, the simulation lattice is too small to sample for the rare events in the percolation path, that in the narrowest point are larger than a given diameter (this was especially true for permeating particle sizes over 20 pixels (0.5μm), where the transport for some cases stopped completely). 

The conditions to find the range of the First Passage Time values for the self-similar scaling was chosen in the following way: from bottom by analyzing the pore diameter distributions (Figure 6) and finding where the power-law scaling breaks down; from top by the condition that the First Passage Time cannot grow more than twofold by increment of the particle size by 1 pixel (26.7 nm). Otherwise, the growth is too large and likely to be caused by limited pore sampling over the finite simulation domain.

The analysis resulted in the First Passage Time ratios for the simulated particles with a diameter varying by a factor of 2 (as with EtOH and H_2_O) within the region of self-similar scaling (second row in Table 2).

These First Passage Time values relate directly to the diffusion coefficients by:(11)$$FPT=\frac{L^{2}}{6\cdot D}$$

However, the established ratio of diffusion coefficients assumes that the jump length and time step for the random walk simulation of the particles with varying sizes are the same. This is not the case; a larger particle receives more collisions, which decreases the time between collisions and the length of the mean free path. To account for this we have scaled the diffusion coefficient ratio predicted by the First Passage Time analysis by the ratio of diffusion coefficients of a water molecule and an ethanol molecule diffusing through an ethanol environment. This translates into changes of the time step between collisions. The ratio of such coefficients can be obtained using the Gilliland formula [38], providing a scaling factor of D_H2O_/D_EtOH_ =1.9, in bulk and in absence of the porous structure:(12)$$D_{1,2}=\frac{0.0043T^{3/2}{\left(\frac{1}{M_{1}}+\frac{1}{M_{2}}\right)}^{1/2}}{P{\left(V_{b1}^{1/3}+V_{b2}^{1/3}\right)}^{2}}$$
where: *D*_1,2_—the diffusion coefficient of species 1 (H_2_O or EtOH, respectively) in species 2 (EtOH); *T*—temperature; *M*—molecular weight of given species; *P*—pressure; and *V*—molal volume.

The resulting values of the effective diffusion coefficient ratios are shown in Table 2 (bottom row), which fit in the range of the experimentally obtained values.

In addition, as given in the experimental results, the CS and CS-P membranes display lower selectivity than the CS-GA and CS-G membranes. Given that the possible errors in the image analysis (binarization, pore statistics), the arbitrary cut-off condition for the self-similarity scaling for large pores, this result is very good and indicates the correct understanding of the transport phenomena in the considered membranes.

### 3.4. Comparison of Experimental Data with Characteristic Structure and Transport Model

Figure 8 compares the intercorrelations between the experimental data and the results obtained from the fractal and random walk analysis. The bar chart shows the ratio of experimental and theoretical water/ethanol diffusion coefficients calculated for an alginate membrane filled with the four various chitosan particles; neat (CS), glycidol-modified (CS-G), glutaraldehyde crosslinked (CS-GA), and phosphorylated (CS-P). The relative errors between the experimental and the modeling results are between 4% and 11%. The smallest and the same value (i.e., 14) of these parameters are obtained for Alg membranes loaded with CS and CS-P particles. It means that the diffusion of water and ethanol particles through the investigated membranes is not very selective, so taking into account only this parameter, Alg membranes with CS and CS-P particles have the worst pervaporation performance. The significantly bigger values of the experimental and theoretical diffusion ratios are observed for membranes with CS-G and CS-GA particles. The ratio in the diffusion coefficients of water and ethanol particles grows as a consequence of the structure of such membranes. Considering the channel length, obtained as a ratio of the average size of the void domain cross-section to the average pore diameter, it can be seen that Alg membranes with CS-G and CS-GA particles are less crowded than membranes with CS and CS-P particles. The longer chains that appear in membranes with chitosan particles modified with glycidol and glutaraldehyde cause fewer obstacles to the movement of the particles along the membrane than in the case of shorter chains. Additionally, the simulation shows that the size of channels which are created in the alginate matrix is not sufficient for easy penetration of ethanol molecules because of their greater size when compared to water molecules. Such a situation favors the separation of water from ethanol. 

## 4. Conclusion

The structure analysis and transport model, based on random walk simulations, were employed to study the transport properties of alginate/chitosan membranes used in the pervaporative dehydration of ethanol. The neat and three modified chitosan particles (glycidol-modified, glutaraldehyde cross-linked, and phosphorylated particles) were used as a filler. Based on membrane images obtained from microscopy, the structural properties of the membranes were characterized by means of the total amount of void space, the average size of void domains, their length and diameter, the fractal dimension, and the degree of self-similarity. The results indicate that despite the use of different chitosan particles, the observed total amount of void space and the values of fractal dimension are very similar. All the investigated membranes display a comparable self-similarity, related to good compatibility of the filling with the polymer matrix. A significant difference is observed in the case of the average size of the void domains. Membranes with CS-GA and CS-G particles are characterized by larger domains than Alg membranes with CS and CS-P particles.

Summing up the results from the modeling, based on the random walk simulations, it can be confirmed that water molecules permeate more easily than ethanol molecules through the membrane because of their smaller size and the availability of more paths formed inside of the membrane. The theoretical results show similar trends to the experimental results received in the pervaporation process. The vapors which permeate through the membranes with CS and CS-P particles display lower diffusion coefficient ratios than in the case of membranes with CS-GA and CS-G particles. Considering the possible errors in the image analysis, the results look very promising and indicate a proper understanding of the transport phenomena in the considered membranes. Furthermore, such research can allow one to consider the effect of the length, diameter, number of channels, and variation in pore diameter on the separation properties of hybrid membranes.

## Figures and Tables

**Figure 1 polymers-12-00411-f001:**
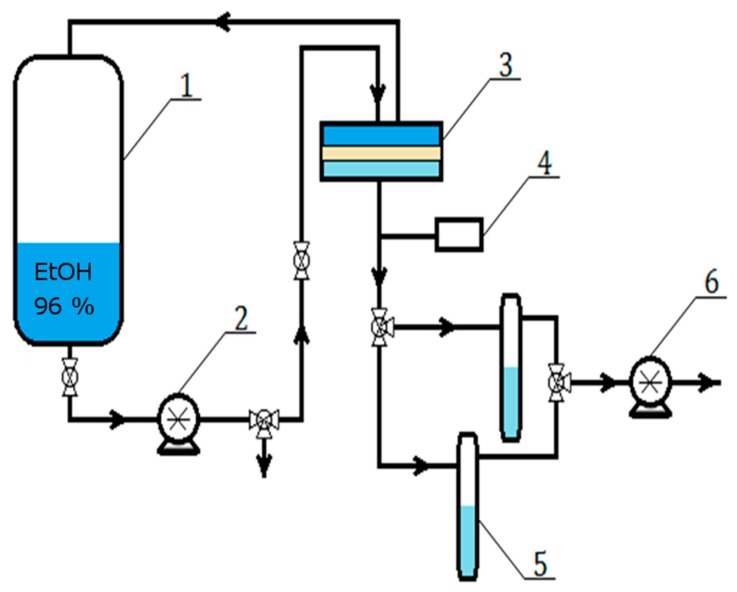
Scheme of the pervaporation setup: 1—feed tank, 2—circulation pump, 3—separation chamber, 4—vacuum gauge, 5—cooled collection traps, and 6—vacuum pump.

**Figure 2 polymers-12-00411-f002:**
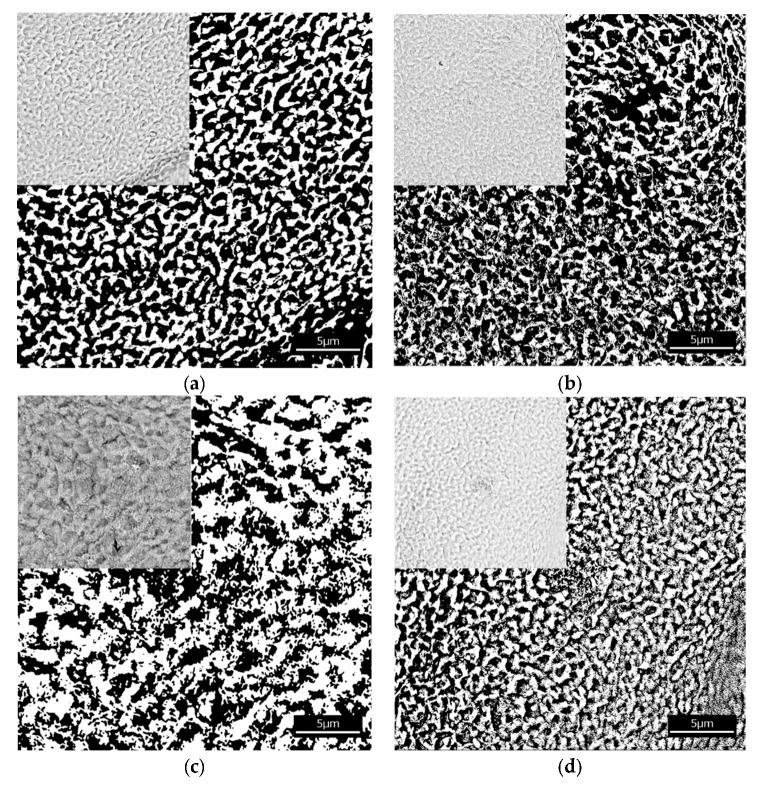
Images of the alginate membranes filled with various chitosan particles (**a**) neat (CS), (**b**) phosphorylated (CS-P), (**c**) glutaraldehyde crosslinked (CS-GA), and (**d**) glycidol-modified (CS-G) both before (upper left side corner) and after binarization. Black regions correspond to the void spaces and white regions to the polymer matrix.

**Figure 3 polymers-12-00411-f003:**
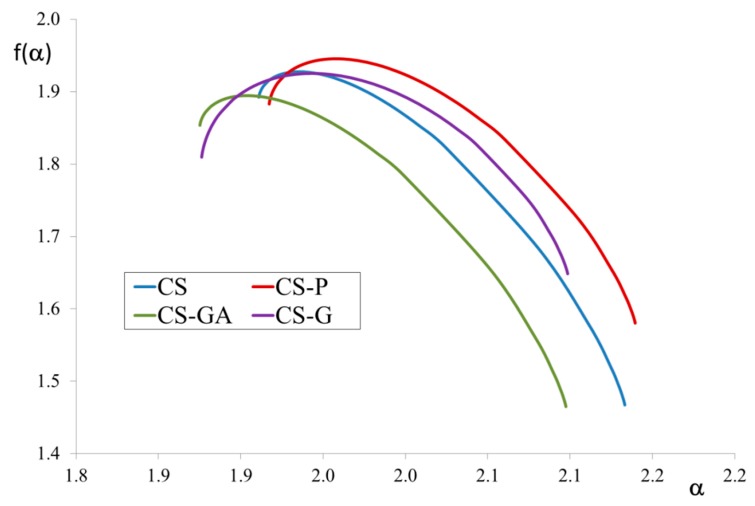
f(α) spectra for alginate/chitosan microparticle membranes filled with chitosan (CS) or modified CS microparticles i.e., phosphorylated chitosan (CS-P), glutaraldehyde crosslinked chitosan (CS-GA), and glycidol-modified chitosan (CS-G).

**Figure 4 polymers-12-00411-f004:**
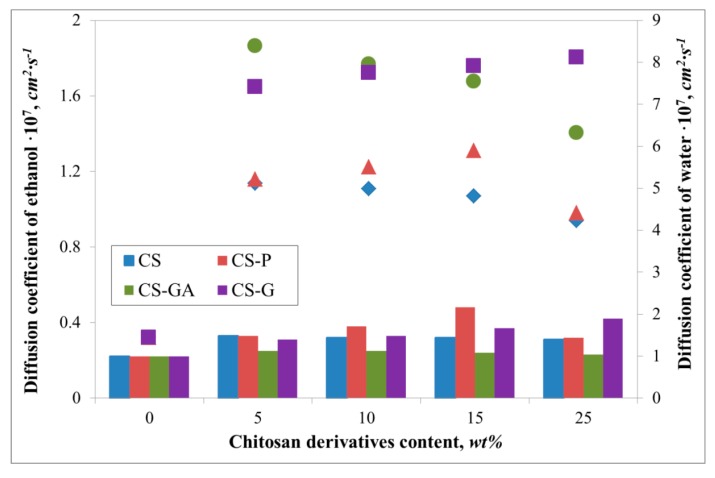
The variation of evaluated diffusion coefficients of water (filled symbols) and ethanol (bar chart) for Alg membranes with increasing content of neat chitosan (CS) and modified CS microparticles; phosphorylated chitosan (CS-P), glutaraldehyde crosslinked chitosan (CS-GA), and glycidol-modified chitosan (CS-G).

**Figure 5 polymers-12-00411-f005:**
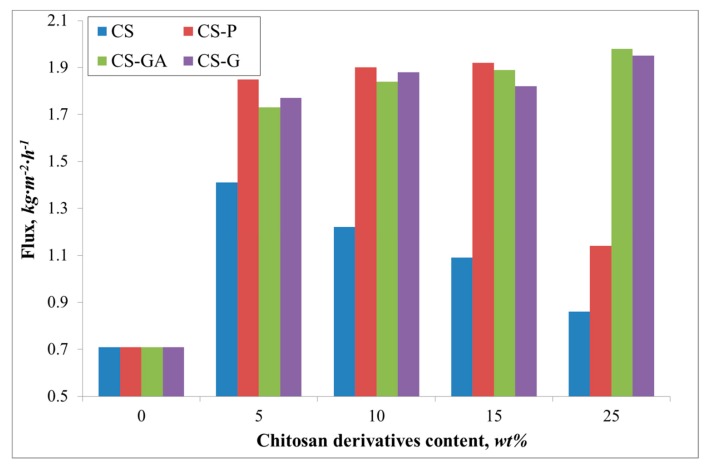
The variation of the measured total fluxes for the Alg membranes with increasing content of chitosan (CS) and modified CS microparticles; phosphorylated chitosan (CS-P), glutaraldehyde crosslinked chitosan (CS-GA), and glycidol-modified chitosan (CS-G).

**Figure 6 polymers-12-00411-f006:**
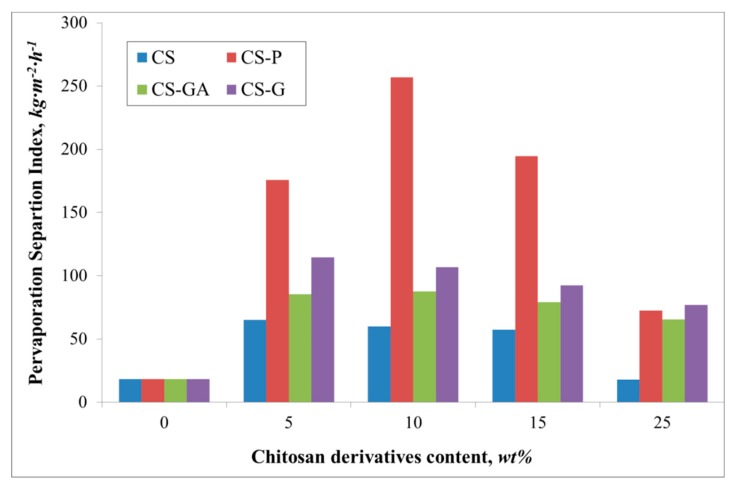
The variation of the measured total Pervaporation Separation Index for Alg membranes with increasing content of chitosan (CS) and modified CS microparticles; phosphorylated chitosan (CS-P), glutaraldehyde crosslinked chitosan (CS-GA), and glycidol-modified chitosan (CS-G).

**Figure 7 polymers-12-00411-f007:**
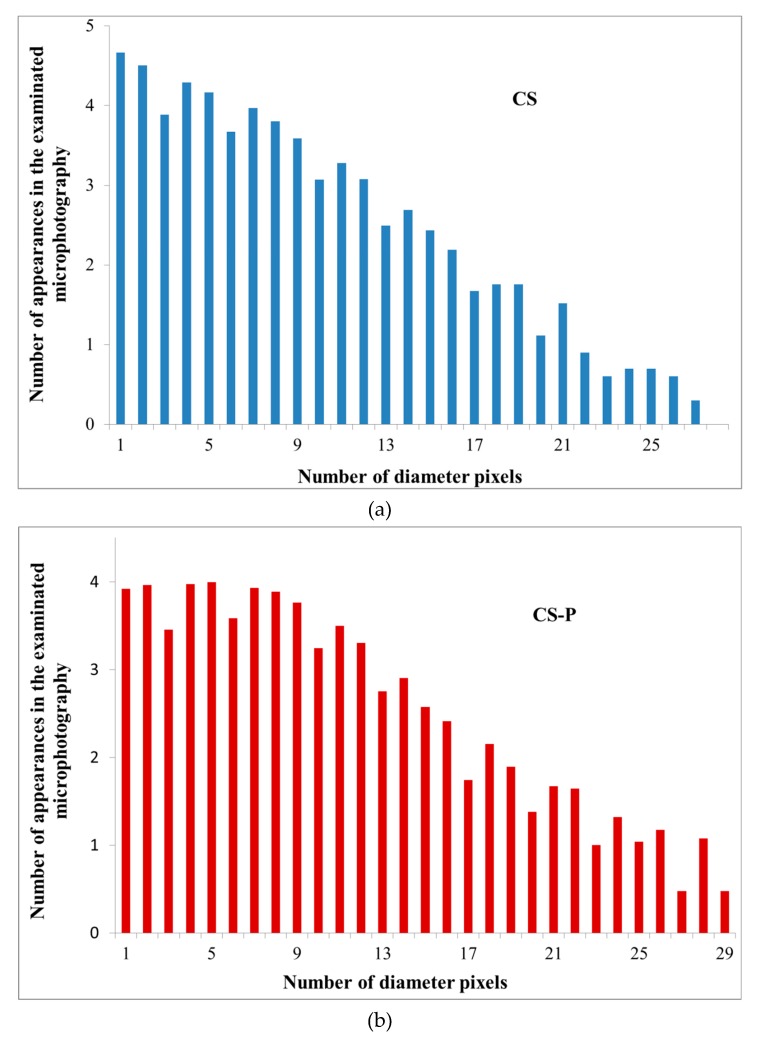
The analysis results of the void diameter distributions for Alg membranes filled with various chitosan particles (**a**) neat (CS), (**b**) phosphorylated (CS-P), (**c**) glutaraldehyde crosslinked (CS-GA), and (**d**) glycidol-modified (CS-G).

**Figure 8 polymers-12-00411-f008:**
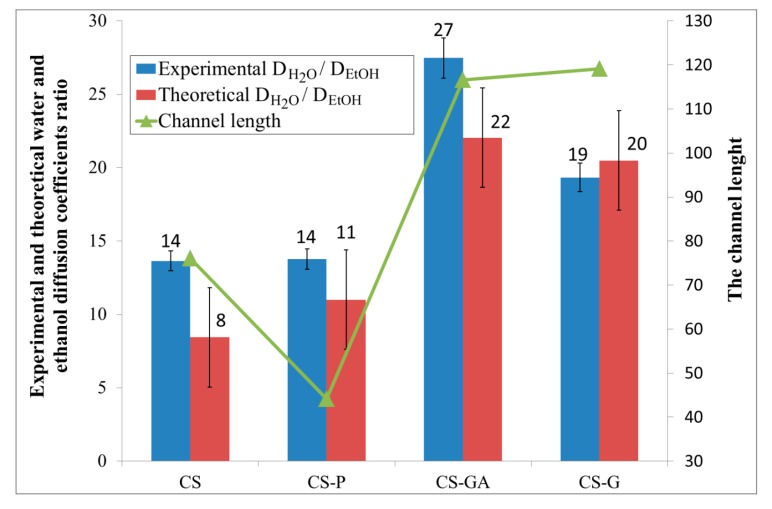
Comparison of the intercorrelations between the experimental data and the results obtained from the fractal and random walk analysis of the alginate membranes filled with neat chitosan (CS) and modified CS microparticles; phosphorylated chitosan (CS-P), glutaraldehyde crosslinked chitosan (CS-GA), and glycidol-modified chitosan (CS-G).

**Table 1 polymers-12-00411-t001:** Structural properties of alginate membranes filled with various chitosan particles.

Chitosan Particles Filling Alginate Membranes	The Observed Total Fraction of Void Space (Porosity) (%)	The Average Size of Void Domains (Pixels)	$\mathrm{Fractal}\;\mathrm{Dimension}\;D_{F}$	Degree of Self-Similarity Δα
Neat (CS)	57	1825	1.93 ± 0.05	0.22
phosphorylated (CS-P)	55	1148	1.96 ± 0.05	0.22
glutaraldehyde crosslinked (CS-GA)	64	4197	1.89 ± 0.05	0.22
glycidol-modified (CS-G)	54	5717	1.94 ± 0.05	0.22

**Table 2 polymers-12-00411-t002:** The First Passage Time ratios and the effective diffusion coefficient ratios for alginate membranes filled with neat chitosan (CS) and modified CS microparticles; phosphorylated chitosan (CS-P), glycidol-modified chitosan (CS-G), and glutaraldehyde crosslinked chitosan (CS-GA).

Particles Parameters	CS	CS-P	CS-GA	CS-G
First Passage Time ratios $FPT_{EtOH}/FPT_{H_{2}O}$	4.4	5.8	11.6	10.8
Effective diffusion coefficient ratios $D_{H_{2}O}/D_{EtOH}$	8.4	11.0	22.0	20.5

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
