# Peer review of "Characterization of the Structure and Transport Properties of Alginate/Chitosan Microparticle Membranes Utilized in the Pervaporative Dehydration of Ethanol"

_polymers, 2020, doi:10.3390/polym12020411_

Round 1

Reviewer 1 Report

Please check the writing carefully in the paper such as Lines 165-170. The quality of the figures is very poor. The format of the figures should be unified. In Fig. 8, 2 should be the subscript.  Please revise the references.

Author Response

Answers to the Reviewer 1 Comments:

The reviewer states that: Please check the writing carefully in the paper such as Lines 165-170.

Answer: We are grateful to Reviewer for careful and thorough review. We have read carefully lines 165-170 and we made appropriate corrections. All corrections made to the text are marked in red.

The reviewer states that: The quality of the figures is very poor. The format of the figures should be unified. In Fig. 8, 2 should be the subscript.

Answer: As suggested, we have improved the figures indicated by the Reviewer.

The reviewer states that: Please revise the references.

Answer: We have read carefully the references and we made appropriate corrections.

Reviewer 2 Report

This manuscript has improved considerably, but there are still many things to be fixed.

1) The quality of figures and tables should be improved further; No figure numbers, spacing, period, symbols, interval, spell error etc

2) The abbreviations are being repeated over and over again (CS, CS-P, CS-G, CS-GA),  are being used inconsistently (Ag_CS, Ag_CS-P, Ag_CS-G, Ag_CS-GA,) or come out without explanation (Alg) in text and figures.

3) There are several sentences whose meanings are hard to get; 74-75, 344-346.

4) In the part of experimental 2.3., font sizes are changing, symbols are missing.  There are some mistakes in space writing in text.

Author Response

Answers to the Reviewer 2 Comments:

The reviewer states that: This manuscript has improved considerably, but there are still many things to be fixed.

The quality of figures and tables should be improved further; No figure numbers, spacing, period, symbols, interval, spell error etc.

Answer: We are grateful to Reviewer for careful and thorough review. As suggested, we have improved the figures and tables indicated by the Reviewer

The abbreviations are being repeated over and over again (CS, CS-P, CS-G, CS-GA),  are being used inconsistently (Ag_CS, Ag_CS-P, Ag_CS-G, Ag_CS-GA,) or come out without explanation (Alg) in text and figures.

Answer: We have read carefully whole article, and we made the corrections that concern abbreviations. We hope that now the article is more readable.

There are several sentences whose meanings are hard to get; 74-75, 344-346

Answer: We have read carefully indicated sentences.

Lines 74-75: was: “studied (…) in the literature” corrected to:  “available (…) in the literature”

Lines 344-346: was: “Therefore, lower values of the separation factor are obtained that result in a smaller Pervaporation Separation Index than for Alg membranes filled with CS-P particles.” corrected to: “Therefore a smaller PSI is obtained, compared to the Alg membranes filled with CS-P particles.”

All corrections made to the text are marked in red.

In the part of experimental 2.3., font sizes are changing, symbols are missing.  There are some mistakes in space writing in text.

Answer: We have read carefully part of experimental 2.3 and we made appropriate corrections. All corrections made to the text are marked in red.

Reviewer 3 Report

The manuscript has been formatted from the previous version. The current article is acceptable.

Author Response

Answers to the Reviewer 3 Comments:

The reviewer states that: The manuscript has been formatted from the previous version. The current article is acceptable.

Answer: We are grateful to Reviewer for reviewing our manuscript and his acceptance for publication.

This manuscript is a resubmission of an earlier submission. The following is a list of the peer review reports and author responses from that submission.

Round 1

Reviewer 1 Report

Dudek et al present a characterization and modeling study pertinent to membranes used in pervaporation, a very important problem for industrial applications. The design and models utilized here in the paper shed insights into how chitosan variants of membranes are optimized for flow of permeates. The investigations presented in this study are probably useful for large scale industrial applications. Based only on structural considerations the authors demonstrated and concluded the chitosan GA/G variants aid better transport compared to CS and CS/P.

The article is acceptable but for few many edits that I think are necessary.

(1) It is requested the authors rectify the grammar in the text for improved readability.

(2) How is the negative sign in Equation 9 coming into picture? Is L_a always negative? Some explanations are appreciated.

(3) Line 316 & 339 the double inverted commas should be taken care of

(4) It is suggested the D_1/2 label be consistent where ever applicable.

(5) Figure 5 the axes labels are missing

(6) Figure 6 the channel length units are missing

(7) The chemical formula for water in in table2 should include proper subscripts

Author Response

Answers to the Reviewer 1

It is requested the authors rectify the grammar in the text for improved readability

Answer: According to the Reviewer suggestion we correct the grammar in the text.

How is the negative sign in Equation 9 coming into picture? Is L_a always negative? Some explanations are appreciated.

Answer: La is negative in case of a non-empty membrane, where vacuum is applied to the permeation side. There is then a rapid desorption, which results in a steep growth of the mass with respect to time. If the initial growth is steeper than the stationary growth of output mass, then the time lag is negative.

Line 316 & 339 the double inverted commas should be taken care of

Answer: According to the Reviewer suggestion we rewrote the paragraph from line 316 to 339.

It is suggested the D_1/2 label be consistent where ever applicable.

Answer: According to the Reviewer suggestion the description of the Gilililand formula has been corrected

Figure 5 the axes labels are missing

Answer: We added the axes labels in Figure 5 (now 6)

Figure 6 the channel length units are missing

Answer: We added the channel length units in Figure 6 (now 7)

The chemical formula for water in in table2 should include proper subscripts

Answer: We corrected the chemical formula for water in table 2

Reviewer 2 Report

For the general reader, you'd better mention briefly the characteristics and importance of alginate/chitosan microparticle membranes and why you choose them and their derivatives for this experiment in the section of introduction. English in this paper is not easy to understand and there are many awkward expressions (10-12, 48, 65, 232, 306, 319, 320, 336, 374). The exclamation marks in the paper look inappropriate. All words and abbreviations (CS, PV, etc) should be marked together when they first come out, and then only abbreviations. You repeat them over and over again every time they come out in your text. However, the abbreviations shown in the figures should be described in figure legends and tables. There are many mistakes in space writing (double space bars in several places, a space bar between the numbers and units, between Fig. and the number, 411, etc). You need to check meticulously and correct them. The quality of the graphs needs to be improved with a bit of sophistication and to be understood at a glance. They look sloppy. A more detailed description is required in figure legends. There is no consistency in the graphs in terms of size, colors, fonts, scales, bullets, titles of X and Y axis, locations and formats of legends in graphs, etc. Even, in the numbers on the scale of some graphs, decimal points are misrepresented with commas. Table is also need to be improved with a more refinement and detailed explanation. I don’t get the meaning of 0,22 in fifth column. Spacing in the front and back of ± and the description of the abbreviations are required. You need to explain or discuss a possible reason for the considerable difference between the calculations and the experimental values in case of CS and CS-GA compared to the others in the section of results and discussions. Conclusion is too long. It needs to be more structured and address the importance and future application of your findings. Several references use an inconsistent format (467, 506, 521, 529). Title is missing in ref 31.

Author Response

Answers to the Reviewer 2

For the general reader, you'd better mention briefly the characteristics and importance of alginate/chitosan microparticle membranes and why you choose them and their derivatives for this experiment in the section of introduction.

Answer: According to the Reviewer suggestion we added the information about alginate/chitosan microparticles membranes at the end of the introduction.

English in this paper is not easy to understand and there are many awkward expressions (10-12, 48, 65, 232, 306, 319, 320, 336, 374). The exclamation marks in the paper look inappropriate.

Answer: According to the Reviewer suggestion we corrected the indicated paragraphs of our manuscript and removed the exclamations marks from the paper.

All words and abbreviations (CS, PV, etc) should be marked together when they first come out, and then only abbreviations. You repeat them over and over again every time they come out in your text. However, the abbreviations shown in the figures should be described in figure legends and tables.

Answer: According to the Reviewer suggestion all words and abbreviations we marked together when they first come out. Additionally we described the abbraviations in the figures and tables label.

There are many mistakes in space writing (double space bars in several places, a space bar between the numbers and units, between Fig. and the number, 411, etc). You need to check meticulously and correct them.

Answer: According to the Reviewer suggestion we checked the whole manuscript and corrected the mistakes. The double spaces and Fig.Number were searched for automatically.

The quality of the graphs needs to be improved with a bit of sophistication and to be understood at a glance. They look sloppy. A more detailed description is required in figure legends. There is no consistency in the graphs in terms of size, colors, fonts, scales, bullets, titles of X and Y axis, locations and formats of legends in graphs, etc. Even, in the numbers on the scale of some graphs, decimal points are misrepresented with commas. Table is also need to be improved with a more refinement and detailed explanation. I don’t get the meaning of 0,22 in fifth column. Spacing in the front and back of ± and the description of the abbreviations are required.

Answer: According to the Reviewer suggestion we improved the quality of the Figures and figures and tables labels. We added the titles of X and Y axis and units.

You need to explain or discuss a possible reason for the considerable difference between the calculations and the experimental values in case of CS and CS-GA compared to the others in the section of results and discussions. Conclusion is too long. It needs to be more structured and address the importance and future application of your findings.

Answer: Differences in the model result from inaccuracies in estimating the scaling of pores according to power law - detection of the least disturbed fragment of pore distribution we have in microscopic data. Appropriate paragraph that describes this problem is provided in the section of results and discussion. Furthermore, we rewrote the conclusion, where we highlight this once again.

Several references use an inconsistent format (467, 506, 521, 529). Title is missing in ref 31.

Answer: We corrected the format of references.

Reviewer 3 Report

The authors studied the structure and transport properties of alginate/chitosan microparticle membranes used in pervaporative dehydration of ethanol and the experimental results were compared with the structure analysis and transport model. Membrane characterization was carried out using images obtained from Phenom SEM microscope. The transport of ethanol and water particles through investigated membranes was modelled using the random walk framework. The review comments for this paper are in the following:

Abstract:

Please add some data from the paper in Abstract and add one sentence about the future application or indications from the current work in the end of Abstract.

Introduction:

What are the main advances in the pervaporation method compared with other methods? L36, what is "PSI"? For the three different classes models in pervaporation, you may need to add more previous works in these paragraphs to illustrate the developments of the mass transfer models. L73, change "extends" to "extended". 

Experimental:

Why did you use 1.5 wt% sodium alginate solution? Especially for a low concentration of the solution? L93, change "concentration" to "concentrations"; change "5; 10; 15; 25 wt%" to "5, 10, 15 and 25 wt%". How could you cast the solutions on the dishes? L103, add "." after "[15]". L106, please unify "h" and "hour". In Section 2.2, please try to use the same concentration unit for the solutions. L115, change "was" to "were". L134, please double check the sentence. L135, it is hard to understand the writing of the paragraph. In Section 2.4, a schematic of the experiment set-up should be given. What are the improvements of the system compared with the one reported in Ref. [24]? L161, please delete the spacing before "where". In Eq. (10), please give the physical meaning of the parameters in this equation. 

Results and discussion:

In Figure 1, what are the porosity of the membranes? Please number all the figures. The quality of the figures needs to be improved. What does "0,22" mean in Table 1? All the figures need to be revised and improved. They are in poor quality. It is hard to tell these data points in Figure 3. Please check the data at the modified chitosan content of 0 wt%. In Figure 4, why did the flux of the membrane containing CS-P decrease at the 25 wt% chitosan content significantly? L328, the paragraph should be revised such as self-similarity!, etc. What are the relative errors between the experimental and the modeling results? More comparisons should be added regarding different parameters.

Conclusion:

Please provide the suitable ranges of the parameters in these models.

Please revise other sections before submission.

Author Response

Abstract: Please add some data from the paper in Abstract and add one sentence about the future application or indications from the current work in the end of Abstract.

Answer: According to the Reviewer suggestion Abstract was rewritten and additional information about the application of describing method was added  .

Introduction: What are the main advances in the pervaporation method compared with other methods? L36, what is "PSI"? For the three different classes models in pervaporation, you may need to add more previous works in these paragraphs to illustrate the developments of the mass transfer models. L73, change "extends" to "extended".

Answer: The main advance in the pervaporation method compared with other method is the lower cost of mixture separation compared with other methods used for separation near boiling mixtures. Additionally such method does not require the use of additional substances which contribute to the contamination of the mixture.

According to the Reviewer suggestion we rewrote the Introduction and corrected mistakes  .

Experimental: Why did you use 1.5 wt% sodium alginate solution? Especially for a low concentration of the solution? L93, change "concentration" to "concentrations"; change "5; 10; 15; 25 wt%" to "5, 10, 15 and 25 wt%". How could you cast the solutions on the dishes? L103, add "." after "[15]". L106, please unify "h" and "hour". In Section 2.2, please try to use the same concentration unit for the solutions. L115, change "was" to "were". L134, please double check the sentence. L135, it is hard to understand the writing of the paragraph. In Section 2.4, a schematic of the experiment set-up should be given. What are the improvements of the system compared with the one reported in Ref. [24]? L161, please delete the spacing before "where". In Eq. (10), please give the physical meaning of the parameters in this equation..

Answer: We used 1.5 wt% sodium alginate solution because if the concentration is too low, the membrane will not be strong enough. The strength depends on the molecular weight, calcium concentration and the grade of alginate.

According to Reviewer suggestion we corrected all mistakes in Experimental Section. We added the experimental set-up as a Figure 1. The set-up was improved by changed the way of collecting the permeate. In the old system, permeate was received in a non-continuous way and in the new one, in a continuous way  .

In Figure 1, what are the porosity of the membranes? Please number all the figures. The quality of the figures needs to be improved. What does "0,22" mean in Table 1? All the figures need to be revised and improved. They are in poor quality. It is hard to tell these data points in Figure 3. Please check the data at the modified chitosan content of 0 wt%. In Figure 4, why did the flux of the membrane containing CS-P decrease at the 25 wt% chitosan content significantly? L328, the paragraph should be revised such as self-similarity!, etc. What are the relative errors between the experimental and the modeling results? More comparisons should be added regarding different parameters.

Answer: In Figure 2 the porosity of the membrane is the number of "black pixels" divided by the number of all pixels. It is actually given in row 2 of table 1.

According to the Reviewer suggestion we improved the quality of all Figures, eg. in Fig. 3 (now 4) we use bar charts., we gave the information about the meaning of “0.22” in Table 1.

The flux decrease at 25 wt% was not yet addressed by the modeling. It remains an open question (?).

The relative errors between the experimental and the modeling results are between 4 and 11%.

Conclusion: Please provide the suitable ranges of the parameters in these models. Please revise other sections before submission.

Answer: For the random walk model, a method is given for evaluating the range of diameters in which power law occurs. Other parameters result from measurements.

According to the Reviewer suggestion we rewrote the whole paper and changed grammar and linguistic errors.

Round 2

Reviewer 3 Report

I do not think that the authors have made reasonable revisions. They did not answer most of my questions (point by point). The paper quality and format still need major improvements.